# Doubly-Robust Lasso Bandit

**Gi-Soo Kim**
Department of Statistics
Seoul National University
gisoo1989@snu.ac.kr

**Myunghee Cho Paik**
Department of Statistics
Seoul National University
myungheechopaik@snu.ac.kr

## Abstract

Contextual multi-armed bandit algorithms are widely used in sequential decision tasks such as news article recommendation systems, web page ad placement algorithms, and mobile health. Most of the existing algorithms have regret proportional to a polynomial function of the context dimension, $d$. In many applications however, it is often the case that contexts are high-dimensional with only a sparse subset of size $s_0(\ll d)$ being correlated with the reward. We consider the stochastic linear contextual bandit problem and propose a novel algorithm, namely the Doubly-Robust Lasso Bandit algorithm, which exploits the sparse structure of the regression parameter as in Lasso, while blending the doubly-robust technique used in missing data literature. The high-probability upper bound of the regret incurred by the proposed algorithm does not depend on the number of arms and scales with $\log(d)$ instead of a polynomial function of $d$. The proposed algorithm shows good performance when contexts of different arms are correlated and requires less tuning parameters than existing methods.

## 1 Introduction

Many sequential decision problems can be framed as the multi-armed bandit (MAB) problem (Robbins, 1952; Lai and Robbins, 1985), where a learner sequentially pulls arms and receives random rewards that possibly differ by arms. While the reward compensation mechanism is unknown, the learner can adapt his (her) decision to the past reward feedback so as to maximize the sum of rewards. Since the rewards of the unchosen arms remain unobserved, the learner should carefully balance between "exploitation", pulling the best arm based on information accumulated so far, and "exploration", pulling the arm that will assist in future choices, although it does not seem to be the best option at the moment. Application areas include the mobile healthcare system (Tewari and Murphy, 2017), web page ad placement algorithms (Langford et al., 2008), news article placement algorithms (Li et al., 2010), revenue management (Ferreira et al., 2018), marketing (Schwartz et al., 2017), and recommendation systems (Kawale et al., 2015).

Contextual MAB algorithms make use of side information, called context, given in the form of finite-dimensional covariates. For example, in the news article recommendation example, information on the visiting user as well as the articles are given in the form of a context vector. In 2010, the Yahoo! team (Li et al., 2010) proposed a contextual MAB algorithm which assumes a linear relationship between the context and reward of each arm. This algorithm achieved a 12.5% click lift compared to a context-free MAB algorithm. Many other bandit algorithms have been proposed for linear reward models (Auer, 2002; Dani et al., 2008; Chu et al., 2011; Abbasi-Yadkori et al., 2011; Agrawal and Goyal, 2013).

The aforementioned methods require the dimension of the context not be too large. This is because in these algorithms, the cumulative gap between the rewards of the optimal arm and the chosen arm, namely regret, is shown to be proportional to a polynomial function of the dimension of the context, $d$. In modern applications however, it is often the case that the web or mobile-based contextual variables

are high-dimensional. Li et al. (2010) applied a dimension reduction method (Chu et al., 2009) to the context variables before applying their bandit algorithm to the Yahoo! news article recommendation log data.

In this paper, we introduce a novel approach to the case where the contextual variables are high-dimensional but only a sparse subset of size $s_0$ is correlated with the reward. We specifically consider the stochastic linear contextual bandit (LCB) problem, where the $N$ arms are represented by $N$ different contexts $b_1(t), \cdots, b_N(t) \in \mathbb{R}^d$ at a specific time $t$, and the rewards $r_1(t), \cdots, r_N(t) \in \mathbb{R}$ are controlled by a single linear regression parameter $\beta \in \mathbb{R}^d$, i.e., $r_i(t)$ has mean $b_i(t)^T \beta$ for $i = 1, \cdots, N$. The LCB problem is distinguished from the contextual bandit problem with linear rewards (CBL), where the context $b(t) \in \mathbb{R}^d$ does not differ by arms but the the reward of the $i$-th arm is determined by an arm-specific parameter $\beta_i \in \mathbb{R}^d$, i.e., $r_i(t)$ has mean $b(t)^T \beta_i$ for $i = 1, \cdots, N$.

When the number of arms is large, the CBL approach is not practical due to large number of parameters. Methods for CBL are also not suited to handle the case where the set of arms changes with time. When we recommend a news article or place an advertisement on the web page, the lists of news articles or advertisements change day by day. In this case, it would not be feasible to assign a different parameter for every new incoming item. Therefore, when the number of arms is large and the set of arms changes with time, LCB approaches including the proposed method can be applied while the CBL approaches cannot.

In supervised learning, Lasso (Tibshirani, 1996) is a good tool for estimating the linear regression parameter when the covariates are high-dimensional but only a sparse subset is related to the outcome. However, the fast convergence property of Lasso is guaranteed when data are i.i.d. and when the observed covariates are not highly correlated, the latter referred to as the compatibility condition (van de Geer and Bühlmann, 2009). In the contextual bandit setting, these conditions are often violated because the observations are adapted to the past and the context variables for which the rewards are observed converge to a small region of the whole context distribution as the learner updates its arm selection policy.

In the proposed method, we resolve the non-compatibility issue by coalescing the methods from missing data literature. We start from the observation that the bandit problem is a missing data problem since the rewards for the arms that are not chosen are not observed, hence, missing. The difference is that in missing data settings, missingness is controlled by the environment and given to the learner while in bandit settings, the missingness is controlled by the learner. Since the learner controls the missingness, the missing mechanism, or arm selection probability is known in bandit settings. Given this probability, we can construct unbiased estimates of the rewards for the arms not chosen, using the doubly-robust technique (Bang and Robins, 2005), which allows capitalizing on the context information corresponding to the arms that are not chosen. These data are observed and available, however, were not utilized by most of the existing contextual bandit algorithms.

We propose the Doubly-Robust Lasso Bandit algorithm which hinges on the sparse structure as in Lasso (Tibshirani, 1996), but utilizes contexts for unchosen arms by blending the doubly-robust technique Bang and Robins (2005) used in missing data literature. The high-probability upper bound of the regret incurred by the proposed algorithm has order $O(s_0 \log(dT)\sqrt{T})$ where $T$ is the total number of time steps. We note that the regret does not depend on the number of arms, $N$, and scales with $\log(d)$ instead of a polynomial function of $d$. Therefore, the regret of the proposed algorithm is sublinear in $T$ even when $N \gg T$ and $d$ scales with $T$.

Abbasi-Yadkori et al. (2012), Carpentier and Munos (2012) and Gilton and Willett (2017) considered the sparse LCB problem as well. While the high-probability regret bound of Abbasi-Yadkori et al. (2012) does not depend on $N$, it is proportional to $\sqrt{d}$ instead of $\log d$, so is not sublinear in $T$ when $d$ scales with $T$. Carpentier and Munos (2012) used an explicit exploration phase to identify the support of the regression parameter using techniques from compressed sensing. Their regret bound is tight scaling with $\log d$, but the algorithm is specific to the case where the set of arms is the unit ball for the $|| \cdot ||_2$ norm and fixed over time. Gilton and Willett (2017) leveraged ideas from linear Thompson Sampling and Relevance Vector Machines (Tipping, 2001). The theoretical results of Gilton and Willett (2017) are weak since they derived the regret bound under the assumption that a sufficiently small superset of the support for the regression parameter is known in advance.

Bastani and Bayati (2015) addressed the CBL problem with high-dimensional contexts. They proposed the Lasso Bandit algorithm which uses Lasso to estimate the parameter of each arm

separately. To solve non-compatibility, Bastani and Bayati (2015) used forced-sampling of each arm at predefined time points and maintained two estimators for each arm, one based on the forced-samples and the other based on all samples. They derived a regret bound of order $O\big(Ns_0^2[\log T + \log d]^2\big)$ but in terms of expectation rather than high-probability. An application of the Hoeffding's inequality would give an additional term of order $O(\sqrt{T})$ for the high-probability bound. For the same problem, Wang et al. (2018) proposed the Minimax Concave Penalized (MCP) Bandit algorithm which uses forced-sampling along with the MCP estimator (Zhang, 2010) and improved the bound of Bastani and Bayati (2015) to $O\big(Ns_0^2[s_0 + \log d]\log T\big)$.

The regrets of Bastani and Bayati (2015) and Wang et al. (2018) increase linearly with $N$ due to separate estimation for each arm. When the number of arms $N$ is bigger than $T$, these algorithms should terminate before the forced-sampling of each arm is completed, thus may not be practical. In contrast, the proposed method allows to share information across arms so its performance does not depend on $N$. In cases where all three methods are applicable, the proposed algorithm requires one less tuning parameter than Lasso Bandit and MCP Bandit, which is a significant advantage for an online learning algorithm as it is difficult to simultaneously tune the hyperparameters and achieve high reward.

We summarize the main contributions of the paper as follows.

- We propose a new linear contextual MAB algorithm for high-dimensional, sparse reward models. The proposed method is simple and requires less tuning parameters than previous works.
- We propose a new estimator for the regression parameter in the reward model which uses Lasso with the context information of all arms through the doubly-robust technique.
- We prove that the high-probability regret upper bound of the proposed algorithm is $O(s_0\log(dT)\sqrt{T})$, which does not depend on $N$ and scales with $\log d$.
- We present experimental results that show the superiority of our method especially when $N$ is big and the contexts of different arms are correlated.

## 1.1 Related Work

The fast convergence of Lasso for linear regression parameter was extenstively studied in van de Geer (2007), Bickel et al. (2009), and van de Geer and Bühlmann (2009). The doubly-robust technique was originally introduced in Robins et al. (1994) and extensively analyzed in Bang and Robins (2005) under supervised learning setup. Dudík et al. (2014), Jiang and Li (2016), and Farajtabar et al. (2018) are recent works that incorporate the doubly-robust methodology into reinforcement learning but under offline learning settings. Dimakopoulou et al. (2018) was the first to blend the doubly-robust technique in the online bandit problem. Their work, which focused on low-dimensional settings, proposed to use the doubly-robust technique as a way of balancing the data over the whole context space which makes the online regression less prone to bias when the reward model is misspecified.

## 2 Settings and Assumptions

In the MAB setting, the learner is repeatedly faced with $N$ alternative arms where at time $t$, the $i$-th arm ($i = 1, \cdots, N$) yields a random reward $r_i(t)$ with unknown mean $\theta_i(t)$. We assume that there is a finite-dimensional context vector $b_i(t) \in \mathbb{R}^d$ associated with each arm $i$ at time $t$ and that the mean of $r_i(t)$ depends on $b_i(t)$, i.e., $\theta_i(t) = \theta_t(b_i(t))$, where $\theta_t(\cdot)$ is an arbitrary function. Among the $N$ arms, the learner pulls one arm $a(t)$, and observes reward $r_{a(t)}(t)$. The optimal arm at time $t$ is $a^*(t) := \underset{1 \le i \le N}{\operatorname{argmax}}\{\theta_t(b_i(t))\}$. Let $regret(t)$ be the difference between the expected reward of the optimal arm and the expected reward of the arm chosen by the learner at time $t$, i.e.,

$$regret(t) = \mathbb{E}\big(r_{a^*(t)}(t) - r_{a(t)}(t) \,\big|\, \{b_i(t)\}_{i=1}^N, a(t)\,\big)$$
$$= \theta_t(b_{a^*(t)}(t)) - \theta_t(b_{a(t)}(t)).$$

Then, the goal of the learner is to minimize the sum of regrets over $T$ steps, $R(T) := \sum_{t=1}^T regret(t)$.

We specifically consider a sparse LCB problem, where $\theta_t(b_i(t))$ is linear in $b_i(t)$,

$$\theta_t(b_i(t)) = b_i(t)^T\beta, \quad i = 1, \cdots, N, \tag{1}$$

where $\beta \in \mathbb{R}^d$ is unknown and $||\beta||_0 = s_0(\ll d)$, $||\cdot||_p$ denoting the $L_p$-norm. Hence, only $s_0$ elements in $\beta$ are assumed to be nonzero. We also make the following assumptions, from A1 to A4.

**A1. Bounded norms.** *Without loss of generality,* $||b_i(t)||_2 \leq 1$, $||\beta||_2 \leq 1$.

**A2. IID Context Assumption.** *The distribution of context variables is i.i.d. over time* $t$:

$$\{b_1(t), \cdots, b_N(t)\} \overset{i.i.d.}{\sim} \mathcal{P}_b,$$

*where* $\mathcal{P}_b$ *is some distribution over* $\mathbb{R}^{N \times d}$.

We note in A2 that given time $t$, the contexts from different arms are allowed to be correlated.

**A3. Compatibility Assumption (van de Geer and Bühlmann, 2009).** *Let* $I$ *be a set of indices and* $\phi$ *be a positive constant. Define*

$$C(I, \phi) = \{M \in \mathbb{R}^{d \times d} : \forall v \in \mathbb{R}^d \text{ such that } ||v_{I^C}||_1 \leq 3||v_I||_1, ||v_I||_1^2 \leq |I|(v^T M v)/\phi^2\}.$$

*Then* $\exists \phi_1 > 0$ *such that*
$$\Sigma := \mathbb{E}[\bar{b}(t)\bar{b}(t)^T] \in C(supp(\beta), \phi_1),$$

*where* $\bar{b}(t) = \frac{1}{N}\sum_{i=1}^N b_i(t)$, *and* $supp(\beta)$ *denotes the support of* $\beta$.

A3 ensures that the Lasso estimate of $\beta$ converges to the true parameter in a fast rate. This assumption is weaker than the restricted eigenvalue assumption (Bickel et al., 2009).

**A4. Sub-Gaussian error.** *The error* $\eta_i(t) := r_i(t) - b_i(t)^T\beta$ *is R-sub-Gaussian for some* $0 < R < O(\sqrt[4]{\log T/T})$, *i.e., for every* $\lambda \in \mathbb{R}$,

$$\mathbb{E}[\exp(\lambda\eta_i(t))] \leq \exp(\lambda^2 R^2/2).$$

Assumption A4 is satisfied whenever $r_i(t) \in [b_i(t)^T\beta - R, b_i(t)^T\beta + R]$.

# 3 Doubly-Robust Lasso Bandit

Lemma 11.2 of van de Geer and Bühlmann (2009) shows that when the covariates satisfy the compatibility condition and the noise is i.i.d. Gaussian, the Lasso estimator of the linear regression parameter converges to the true parameter in a fast rate. We restate the lemma.

**Lemma 3.1. (Lemma 11.2 of van de Geer and Bühlmann, 2009)** *Let* $x_\tau \in \mathbb{R}^d$ *and* $y_\tau \in \mathbb{R}$ *be random variables with* $y_\tau = x_\tau^T\beta + \varepsilon_\tau$, $\tau = 1, 2, \cdots, t$, *where* $\beta \in \mathbb{R}^d$, $||\beta||_0 = s_0$, *and* $\varepsilon_\tau$*'s are i.i.d. Gaussian with mean zero and variance* $R^2$. *Let* $\hat{\beta}(t)$ *be the Lasso estimator of* $\beta$ *based on the first* $t$ *observations using* $\lambda_t = R\sqrt{\frac{2\log(ed/\delta)}{t}}$ *for the regularization parameter in the* $L_1$ *penalty, i.e.,*

$$\hat{\beta}(t) = \underset{\beta}{\operatorname{argmin}}\Big\{\frac{1}{t}\sum_{\tau=1}^t (y_\tau - x_\tau^T\beta)^2 + \lambda_t||\beta||_1\Big\}.$$

*If* $\hat{\Sigma}_t := \frac{1}{t}\sum_{\tau=1}^t x_\tau x_\tau^T \in C(supp(\beta), \phi)$ *for some* $\phi > 0$, *then for* $\forall \delta \in (0, 1)$, *with probability at least* $(1 - \delta)$,

$$||\hat{\beta}(t) - \beta||_1 \leq \frac{4\lambda_t s_0}{\phi^2} = \frac{4s_0 R}{\phi^2}\sqrt{\frac{2\log(ed/\delta)}{t}}. \tag{2}$$

Two hurdles that arise when incorporating Lemma 3.1 into the contextual MAB setting are that (a) (non-i.i.d.) the errors $\varepsilon_\tau$'s are not i.i.d., and (b) (non-compatibility) the Gram matrix $\hat{\Sigma}_t$ does not satisfy the compatibility condition even under assumption A3 because the context variables of which the rewards are observed do not evenly represent the whole distribution of the context variables. In the LCB setting, $x_\tau$ corresponds to $b_{a(\tau)}(\tau)$. Since the learner tends to choose the contexts that yield maximum reward as time elapses, the chosen context variables $b_{a(\tau)}(\tau)$'s converge to a small region of the whole context space. We discuss on remedies to (a) and (b) in Section 3.1 and Section 3.2, respectively.

### 3.1 Lasso Oracle Inequality with non-i.i.d. noise

As a remedy to the non-i.i.d. problem, Bastani and Bayati (2015) proposed a variation of Lemma 3.1 which shows that (2) holds when $\{\varepsilon_\tau\}_{\tau=1}^t$ is a martingale difference sequence. We restate the proposition of Bastani and Bayati (2015).

**Lemma 3.2. (Proposition 1 of Bastani and Bayati, 2015)** *Let $\mathcal{F}_{\tau-1}$ denote the filtration up to time $\tau-1$ in the bandit setting,*

$$\mathcal{F}_{\tau-1} = \{x_1, y_1, \cdots, x_{\tau-1}, y_{\tau-1}, x_\tau\}, \quad \tau = 1, 2, \cdots, t.$$

*Suppose the conditions of Lemma 3.1 hold except that $\varepsilon_\tau$'s are i.i.d. Suppose instead that $\varepsilon_\tau | \mathcal{F}_{\tau-1}$ is $R$-sub-Gaussian for $\tau = 1, \cdots, t$. Then for $\forall \delta \in (0,1)$, with probability at least $(1-\delta)$,*

$$||\hat{\beta}(t) - \beta||_1 \leq \frac{4\lambda_t s_0}{\phi^2} = \frac{4 s_0 R}{\phi^2} \sqrt{\frac{2\log(ed/\delta)}{t}}. \tag{3}$$

### 3.2 Doubly-robust pseudo-reward

To overcome the non-compatibility problem, in the CBL setting, Bastani and Bayati (2015) proposed to impose forced-sampling of each arm at predefined $O(\log T)$ time steps, which produces i.i.d. data. The following lemma shows that when $x_\tau$'s are i.i.d. and $\mathbb{E}(x_\tau x_\tau^T) \in C(supp(\beta), \phi)$, then $\hat{\Sigma}_t \in C(supp(\beta), \phi/\sqrt{2})$ with high probability.

**Lemma 3.3. (Lemma EC.6. of Bastani and Bayati, 2015)** *Let $x_1, x_2, \cdots, x_t$ be i.i.d. random vectors in $\mathbb{R}^d$ with $||x_\tau||_\infty \leq 1$ for all $\tau$. Let $\Sigma = \mathbb{E}[x_\tau x_\tau^T]$ and $\hat{\Sigma}_t = \frac{1}{t} \sum_{\tau=1}^t x_\tau x_\tau^T$. Suppose that $\Sigma \in C(supp(\beta), \phi)$. Then if $t \geq \frac{3}{c^2}\log d$,*

$$\mathbb{P}\Big[\hat{\Sigma}_t \in C\Big(supp(\beta), \frac{\phi}{\sqrt{2}}\Big)\Big] \geq 1 - \exp\big(-c^2 t\big), \tag{4}$$

*where $c = \min(0.5, \frac{\phi^2}{256 s_0})$.*

We derive the following corollary simply by setting the right-hand side of (4) larger than $1 - \frac{\delta'}{t^2}$.

**Corollary 3.4.** *Suppose that the conditions of Lemma 3.3 are satisfied. Let $z_T = \max\big(\frac{3}{c^2}\log d, \frac{1}{c^2}\log\frac{T^2}{\delta'}\big)$ for some $\delta' \in (0,1)$. Then with probability at least $1 - \delta'$,*

$$\hat{\Sigma}_t \in C\Big(supp(\beta), \frac{\phi}{\sqrt{2}}\Big) \text{ for all } t \geq z_T. \tag{5}$$

The setting of Bastani and Bayati (2015) is different from ours in that they assume the context variable is the same for all arms but the regression parameter differs by arms. Bastani and Bayati (2015) maintained two sets of estimators for each arm, the estimator based on the forced-samples and the one based on all samples. Whereas the latter is not based on i.i.d. data, using the forced-sample estimator as a pre-processing step of selecting a subset of arms and then using the all-sample estimator to select the best arm among this subset guarantees that there are $O(T)$ i.i.d. samples for each arm, so the all-sample estimator converges in a fast rate as well.

We propose a different approach to resolve non-compatibility, which is based on the doubly-robust technique in missing data literature. We define the filtration $\mathcal{F}_{t-1}$ as the union of the observations until time $t-1$ and the contexts given at time $t$, i.e.,

$$\mathcal{F}_{t-1} = \{\{b_i(1)\}_{i=1}^N, a(1), r_{a(1)}(1), \cdots, \{b_i(t-1)\}_{i=1}^N, a(t-1), r_{a(t-1)}(t-1), \{b_i(t)\}_{i=1}^N\}.$$

Given $\mathcal{F}_{t-1}$, we pull the arm $a(t)$ randomly according to probability $\pi(t) = [\pi_1(t), \cdots, \pi_N(t)]$, where $\pi_i(t) = \mathbb{P}[a(t) = i | \mathcal{F}_{t-1}]$ is the probability of pulling the $i$-th arm at time $t$ given $\mathcal{F}_{t-1}$. We specify the values of $\pi(t)$ later in Section 3.3. Let $\hat{\beta}(t-1)$ be the $\beta$ estimate given $\mathcal{F}_{t-1}$. After the reward $r_{a(t)}(t)$ is observed, we construct a doubly-robust pseudo-reward:

$$\hat{r}(t) = \bar{b}(t)^T \hat{\beta}(t-1) + \frac{1}{N} \frac{r_{a(t)}(t) - b_{a(t)}(t)^T \hat{\beta}(t-1)}{\pi_{a(t)}(t)}.$$

Whether or not $\hat{\beta}(t-1)$ is a valid estimate, this value has conditional expectation $\bar{b}(t)^T\beta$ given that $\pi_i(t) > 0$ for all $i$:

$$\mathbb{E}[\hat{r}(t)|\mathcal{F}_{t-1}] = \mathbb{E}\Big[\frac{1}{N}\sum_{i=1}^{N}\Big(1 - \frac{I(a(t)=i)}{\pi_i(t)}\Big)b_i(t)^T\hat{\beta}(t-1) + \frac{1}{N}\sum_{i=1}^{N}\frac{I(a(t)=i)}{\pi_i(t)}r_i(t)\Big|\mathcal{F}_{t-1}\Big]$$

$$= \mathbb{E}\Big[\frac{1}{N}\sum_{i=1}^{N}r_i(t)\Big|\mathcal{F}_{t-1}\Big] = \bar{b}(t)^T\beta.$$

Thus by weighting the observed reward $r_{a(t)}(t)$ with the inverse of its observation probability $\pi_{a(t)}(t)$, we obtain an unbiased estimate of the reward corresponding to the average context $\bar{b}(t)$ instead of $b_{a(t)}(t)$. Applying Lasso regression to the pair $(\bar{b}(t), \hat{r}(t))$ instead of $(b_{a(t)}(t), r_{a(t)}(t))$, we can make use of the i.i.d. assumption A2 and the compatibility assumption A3 with Corollary 3.4 to achieve a fast convergence rate for $\beta$ as in (3). We later show that when $x_\tau$ in Corollary 3.4 corresponds to $\bar{b}(\tau)$, (5) holds with $z_T = O(s_0^2\log(dT))$.

Since the allocation probability $\pi_{a(t)}(t)$ is known given $\mathcal{F}_{t-1}$, the simple inverse probability weighting (IPW) estimator $\breve{r}(t) := \frac{r_{a(t)}(t)}{N\pi_{a(t)}(t)}$ also gives an unbiased estimate for the reward corresponding to the average context. We however show in the next paragraphs that the doubly-robust estimator has constant variance under minor condition on the weight $\pi_{a(t)}(t)$ while the IPW estimator has variance that increases with $t$ under the same condition. Constant variance is crucial for the performance of the algorithm, since it affects the convergence property of $\hat{\beta}(t)$ in (3) and eventually the regret upper bound in Theorem 4.1. We can make the IPW estimator have constant variance as well under stronger condition on $\pi_{a(t)}(t)$, but this stronger condition is shown to result in regret linear in $T$.

Let $\tilde{R}^2$ be the variance of $\hat{r}(t)$ given $\mathcal{F}_{t-1}$. We have,

$$\tilde{R}^2 := Var[\hat{r}(t)|\mathcal{F}_{t-1}] = \mathbb{E}\big[\{\hat{r}(t) - \bar{b}(t)^T\beta\}^2\big|\mathcal{F}_{t-1}\big]$$

$$= \mathbb{E}\Big[\Big\{\bar{b}(t)^T\big(\hat{\beta}(t-1) - \beta\big) + \frac{\eta_{a(t)}(t)}{N\pi_{a(t)}(t)} + \frac{b_{a(t)}(t)^T(\beta - \hat{\beta}(t-1))}{N\pi_{a(t)}(t)}\Big\}^2\Big|\mathcal{F}_{t-1}\Big]. \qquad (6)$$

Suppose $\hat{\beta}(t-1)$ satisfies the Lasso convergence property (3), i.e., $\|\hat{\beta}(t-1) - \beta\|_1 \leq O(\sqrt{(\log d + \log t)/t})$ for $t > z_T$. Then we can bound the first and third terms of (6) by a constant under the following restriction,

$$\pi_{a(t)}(t) = \begin{cases} \frac{1}{N} & \text{for all } t \leq z_T \\ O\big(\frac{1}{N}\sqrt{(\log d + \log t)/t}\big) & \text{for all } t > z_T. \end{cases} \qquad (7)$$

Due to assumption A4 on the sub-Gaussian error $\eta$, the second term of (6) is also bounded by a constant under (7). Hence, we have $\tilde{R}^2 \leq O(1)$. Therefore if (7) holds and if $\hat{\beta}(z_T), \hat{\beta}(z_T + 1), \cdots, \hat{\beta}(t-1)$ all satisfy the Lasso convergence property (3), then the pseudo-rewards $\hat{r}(1), \cdots, \hat{r}(t)$ all have constant variance. Consequently, the estimate $\hat{\beta}(t)$ based on $\{\bar{b}(\tau), \hat{r}(\tau)\}_{\tau=1}^{t}$ satisfies (3). Meanwhile, the restriction (7) leads to suboptimal choice of arms at each $t$ with probability that decreases with time. In Theorem 4.1, we prove that the regret due to this suboptimal choice is sublinear in $T$.

The variance of $\breve{r}(t)$ given filtration $\mathcal{F}_{t-1}$ is $\mathbb{E}\Big[\Big\{\frac{\eta_{a(t)}(t)}{N\pi_{a(t)}(t)} + \frac{b_{a(t)}(t)^T\beta}{N\pi_{a(t)}(t)} - \bar{b}(t)^T\beta\Big\}^2\Big|\mathcal{F}_{t-1}\Big]$. As opposed to the first and third terms in (6), the term $b_{a(t)}(t)^T\beta/N\pi_{a(t)}(t)$ still increases with $t$ under (7). To achieve a constant variance, we need $\pi_{a(t)}(t)$ be larger than a predetermined constant value, $\frac{1}{N}p_{min}$. Simply replacing $\pi_{a(t)}(t)$ in $\breve{r}(t)$ with the truncated value $\breve{\pi}_{a(t)}(t) = \max\{\frac{1}{N}p_{min}, \pi_{a(t)}(t)\}$ will produce a biased estimate of $\bar{b}(t)^T\beta$ when $\pi_{a(t)}(t)$ is actually smaller than $\frac{1}{N}p_{min}$, and the resulting estimate $\hat{\beta}(t)$ will not satisfy convergence property (3). If we instead directly constrain $\pi_{a(t)}(t)$ to be larger than $\frac{1}{N}p_{min}$, this will lead to suboptimal choice of arms at each $t$ with constant probability so the regret will increase linearly in $T$.

### 3.3 Doubly-Robust Lasso Bandit algorithm

To make $\pi_{a(t)}(t) = \frac{1}{N}$ for all $t \leq z_T$, we simply pull arms according to the uniform distribution when $t \leq z_T$. Then to ensure $\pi_{a(t)}(t) \geq O\left(\frac{1}{N}\sqrt{(\log d + \log t)/t}\right)$ for all $t > z_T$, we randomize the arm selection at each step between uniform random selection and deterministic selection using the $\beta$ estimate of the last step, $\hat{\beta}(t-1)$. This can be implemented in two stages. First, sample $m_t$ from a Bernouilli distribution with mean $\lambda_{1t} = \lambda_1\sqrt{(\log t + \log d)/t}$, where $\lambda_1 > 0$ is a tuning parameter. Then if $m_t = 1$, pull the arm $a(t) = i$ with probability $1/N$, otherwise, pull the arm $a(t) = \operatorname*{argmax}_{1 \leq i \leq N}\{b_i(t)^T\hat{\beta}(t-1)\}$. Under this procedure, we have $\pi_{a(t)}(t) = \lambda_{1t}/N + (1 - \lambda_{1t})I\{a(t) = \operatorname*{argmax}_{1 \leq i \leq N}\{b_i(t)^T\hat{\beta}(t-1)\}\}$, which satisfies (7). In practice, we treat $z_T$ as a tuning parameter.

The proposed Doubly-Robust (DR) Lasso Bandit algorithm is presented in Algorithm 1. The algorithm selects $a(t)$ via two stages, computes $\pi_{a(t)}(t)$, constructs the pseudo-reward $\hat{r}(t)$ and average context $\bar{b}(t)$, and updates the $\beta$ estimate using Lasso regression on the pairs $(\bar{b}(t), \hat{r}(t))$'s. Recall that the regularization parameter should be set as $\lambda_{2t} = O\left(\sqrt{\log(d/\delta)/t}\right)$ to guarantee (3) for a fixed $t$. To guarantee (3) for every $t > z_T$, we impute $\delta/t^2$ instead of $\delta$. Hence at time $t$, we update $\lambda_{2t} = \lambda_2\sqrt{(\log t + \log d)/t}$ where $\lambda_2 > 0$ is a tuning parameter. The algorithm requires three tuning parameters in total, while Bastani and Bayati (2015) and Wang et al. (2018) require four.

---

**Algorithm 1** DR Lasso Bandit
***
    Input parameters: $\lambda_1, \lambda_2, z_T$
    Set $\hat{\beta}(0) = 0_d, \mathbb{S} = \{\ \}$.
    **for** $t = 1, 2, \cdots, T$ **do**
        Observe $\{b_1(t), b_2(t), \cdots, b_N(t)\} \sim \mathcal{P}_b$
        **if** $t \leq z_T$ **then**
            Pull arm $a(t) = i$ with probability $\frac{1}{N}$ $(i = 1, \cdots, N)$
            $\pi_{a(t)}(t) \leftarrow 1/N$
        **else**
            $\lambda_{1t} \leftarrow \lambda_1\sqrt{(\log t + \log d)/t}$, sample $m_t \sim Ber(\lambda_{1t})$
            **if** $m_t = 1$ **then**
                Pull arm $a(t) = i$ with probability $\frac{1}{N}$ $(i = 1, \cdots, N)$
            **else**
                Pull arm $a(t) = \operatorname*{argmax}_{1 \leq i \leq N}\{b_i(t)^T\hat{\beta}(t-1)\}$
            **end if**
            $\pi_{a(t)}(t) \leftarrow \lambda_{1t}/N + (1 - \lambda_{1t})I\{a(t) = \operatorname*{argmax}_{1 \leq i \leq N}\{b_i(t)^T\hat{\beta}(t-1)\}\}$
        **end if**
        Observe $r_{a(t)}(t)$
        $\bar{b}(t) \leftarrow \frac{1}{N}\sum_{i=1}^{N} b_i(t)$,   $\hat{r}(t) \leftarrow \bar{b}(t)^T\hat{\beta}(t-1) + \frac{1}{N}\frac{r_{a(t)}(t) - b_{a(t)}(t)^T\hat{\beta}(t-1)}{\pi_{a(t)}(t)}$
        $\mathbb{S} \leftarrow \mathbb{S} \cup \{(\bar{b}(t), \hat{r}(t))\}$
        $\lambda_{2t} \leftarrow \lambda_2\sqrt{(\log t + \log d)/t}$
        $\hat{\beta}(t) \leftarrow \operatorname*{argmin}_{\beta}\{\frac{1}{t}\sum_{(\bar{b},\hat{r}) \in \mathbb{S}}(\hat{r} - \bar{b}^T\beta)^2 + \lambda_{2t}||\beta||_1\}$
    **end for**
***

## 4 Regret analysis

Under (1), the regret at time $t$ is $regret(t) = b_{a^*(t)}(t)^T\beta - b_{a(t)}(t)^T\beta$, where $a^*(t) = \operatorname*{argmax}_{1 \leq i \leq N}\{b_i(t)^T\beta\}$. We establish the high-probability regret upper bound for Algorithm 1 in the following theorem.

**Theorem 4.1.** *Suppose (1) and assumptions A1, A2, A3, and A4 hold. Then we have for $\forall \delta \in (0, \frac{1}{2})$ and $\forall \delta' \in (0, \delta)$, with probability at least $1 - 2\delta$,*

$$R(T) \leq \max\left( \frac{3}{C(\phi_1)^2} \log d, \frac{1}{C(\phi_1)^2} \log\left(\frac{T^2}{\delta'}\right)\right) + \frac{\sqrt{128}}{\phi_1^2} s_0 \tilde{R} \sqrt{T} \log\left(\frac{edT^2}{\delta - \delta'}\right) + \lambda_1 \sqrt{T} \log(dT)$$

$$+ \sqrt{\frac{T}{2} \log\left(\frac{1}{\delta}\right)}$$

$$= O\left(s_0 \sqrt{T} \log(dT)\right),$$

*where $C(\phi_1) = \min(0.5, \frac{\phi_1^2}{256 s_0})$ and $\tilde{R} = O(1)$.*

We provide a sketch of proof in the next section. A complete proof is presented in the Supplementary Material.

## 4.1 Sketch of proof for Theorem 4.1

In the DR Lasso Bandit algorithm, each of the $T$ decision steps corresponds to one of the following three groups.

(a) $t \leq z_T$ : the arms are pulled according to the uniform distribution.

(b) $t > z_T$ and $m_t = 1$ : the arms are pulled according to the uniform distribution.

(c) $t > z_T$ and $m_t = 0$ : the arm $a(t) = \operatorname*{argmax}_{1 \leq i \leq N} \{b_i(t)^T \hat{\beta}(t-1)\}$ is pulled.

We let $z_T = \max\left( \frac{3}{C(\phi_1)^2} \log d, \frac{1}{C(\phi_1)^2} \log\left(\frac{T^2}{\delta'}\right)\right)$. The strategy of the proof is to bound the regrets from each separate group and sum the results.

Due to assumption A1 on the norms of $b_i(t)$ and $\beta$, $regret(t)$ is not larger than 1 in any case. Therefore, the sum of regrets from group (a) is at most $z_T$, which corresponds to the first term in Theorem 4.1. We now denote the sum of regrets from group (b) and group (c) as $R(T, b)$ and $R(T, c)$, respectively. We first bound $R(T, b)$ in Lemma 4.2, which follows from the fact that $m_t$ is a Bernouilli variable with mean $\lambda_{1t}$ and Hoeffding's inequality.

**Lemma 4.2.** *With probability at least $1 - \delta$,*

$$R(T, b) \leq \sum_{t=z_T}^{T} \lambda_{1t} + \sqrt{T \log(1/\delta)/2} \leq \lambda_1 \sqrt{T} \sqrt{\log(dT)} \sqrt{1 + \log T} + \sqrt{T \log(1/\delta)/2}.$$

To bound $R(T, c)$, we further decompose group (c) into the following two subgroups, where we set $d_t = \frac{\sqrt{128}}{\phi_1^2} s_0 \tilde{R} \sqrt{\frac{\log(edt^2/(\delta-\delta'))}{t}}$ with $\tilde{R} = O(1)$.

(c-a) $t > z_T, m_t = 0 : a(t) = \operatorname*{argmax}_{1 \leq i \leq N} \{b_i(t)^T \hat{\beta}(t-1)\}$ is pulled and $||\hat{\beta}(t-1) - \beta||_1 \leq d_{t-1}$.

(c-b) $t > z_T, m_t = 0 : a(t) = \operatorname*{argmax}_{1 \leq i \leq N} \{b_i(t)^T \hat{\beta}(t-1)\}$ is pulled and $||\hat{\beta}(t-1) - \beta||_1 > d_{t-1}$.

We first show in Lemma 4.3 that the subgroup (c-b) is empty with high probability. Since $\hat{\beta}(t)$ is computed from the pairs $\{\bar{b}(\tau), \hat{r}(\tau)\}_{\tau=1}^{t}$ where the average contexts $\bar{b}(\tau)$'s satisfy the compatibility condition (5) and the pseudo-rewards $\hat{r}(\tau)$'s are unbiased with constant variance, Lemma 3.2 can be applied on $\hat{\beta}(t)$.

**Lemma 4.3.** *Suppose (1) and assumptions A1, A2, A3, and A4 hold. Let $d_t = \frac{\sqrt{128}}{\phi_1^2} s_0 \tilde{R} \sqrt{\frac{\log(edt^2/(\delta-\delta'))}{t}}$. Then we have for $\forall \delta \in (0, 1)$, for every $t \geq z_T$,*

$$\mathbb{P}\left( ||\hat{\beta}(t) - \beta||_1 > d_t \,\Big|\, ||\hat{\beta}(t-1) - \beta||_1 < d_{t-1}, \cdots, ||\hat{\beta}(z_T) - \beta||_1 < d_{z_T} \right) \leq \frac{\delta}{t^2}. \quad (8)$$

*Hence, with probability at least $1 - \delta$,*

$$||\hat{\beta}(t) - \beta||_1 < d_t \text{ for every } t \geq z_T.$$

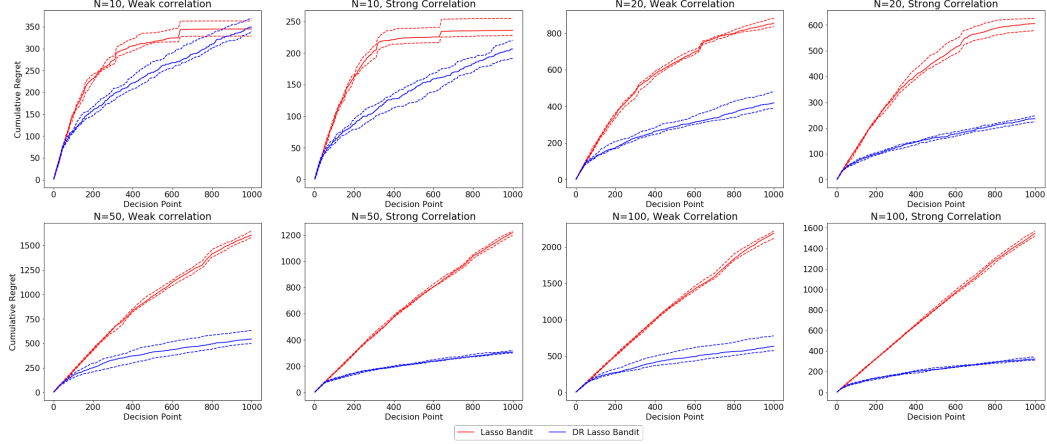

Figure 1: Median (solid), 1st and 3rd quartiles (dashed) of $R(t)$ over 10 experiments.

Finally when $t$ belongs to group (c-a), $regret(t)$ is trivially bounded by $d_t$. Hence we can bound $R(T, c)$ as in the following lemma.

**Lemma 4.4.** *With probability at least* $1 - \delta$,

$$R(T, c) \leq \sum_{t=z_T}^{T} d_t \leq \frac{\sqrt{128}}{\phi_1^2} s_0 \tilde{R} \sqrt{T} \log\Big(\frac{edT^2}{\delta - \delta'}\Big).$$

## 5 Simulation study

We conduct simulation studies to evaluate the proposed DR Lasso Bandit and the Lasso bandit (Bastani and Bayati, 2015). We set $N = 10$, 20, 50, or 100, $d = 100$, and $s_0 = 5$. For fixed $j = 1, \cdots, d$, we generate $[b_{1j}(t), \cdots, b_{Nj}(t)]^T$ from $\mathcal{N}(0_N, V)$ where $V(i, i) = 1$ for every $i$ and $V(i, k) = \rho^2$ for every $i \neq k$. We experiment two cases for $\rho^2$, either $\rho^2 = 0.3$ (weak correlation) or $\rho^2 = 0.7$ (strong correlation). We generate $\eta_i(t) \overset{i.i.d.}{\sim} \mathcal{N}(0, 0.05^2)$ and the rewards from (1). We set $||\beta||_0 = s_0$ and generate the $s_0$ non-zero elements from a uniform distribution on $[0, 1]$.

We conduct 10 replications for each case. The Lasso Bandit algorithm can be applied in our setting by considering a $Nd$-dimensional context vector $b(t) = [b_1(t)^T, \cdots, b_N(t)^T]^T$ and a $Nd$-dimensional regression parameter $\beta_i$ for each arm $i$ where $\beta_i = [I(i = 1)\beta^T, \cdots, I(i = N)\beta^T]^T$. For each algorithm, we consider some candidates for the tuning parameters and report the best results. For DR Lasso Bandit, we advise to truncate the value $\hat{r}(t)$ so that it does not explode.

Figure 1 shows the cumulative regret $R(t)$ according to time $t$. When the number of arms $N$ is as small as 10, Lasso Bandit converges faster to the optimal decision rule than DR Lasso Bandit. However, we notice that the regret of Lasso Bandit in the early stage is larger than DR Lasso Bandit, which is due to forced-sampling of arms. With larger number of arms, DR Lasso Bandit outperforms Lasso Bandit. Compared to Lasso Bandit, the regret of DR Lasso Bandit decreases dramatically as the correlation among the contexts of different arms increases. We also verify that the performance of the proposed method is not sensitive to $N$, while the regret of Lasso Bandit increases with $N$.

## 6 Concluding remarks

Sequential decision problems involving web or mobile-based data call for contextual MAB methods that can handle high-dimensional covariates. In this paper, we propose a novel algorithm which enjoys the convergence properties of the Lasso estimator for i.i.d. data via a doubly-robust technique established in the missing data literature. The proposed algorithm attains a tight high-probability regret upper bound which depends on a polylogarithmic function of the dimension and does not depend on the number of arms, overcoming weaknesses of the existing algorithms.

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
