[Supplementary Material · Doubly_Robust_Lasso_Bandit__Supplementary (1).pdf]

# Supplementary Material:
# Doubly-Robust Lasso Bandit

**Gi-Soo Kim**
Department of Statistics
Seoul National University
gisoo1989@snu.ac.kr

**Myunghee Cho Paik**
Department of Statistics
Seoul National University
myungheechopaik@snu.ac.kr

## A   Proof of Lemma 4.2

Since $regret(t)$ is not larger than 1 in any case, $R(T, b) \leq \sum_{t=z_T}^{T} m_t$, where $\{m_t\}_{t=z_T}^{T}$ is a sequence of independent Bernoulli variables with $\mathbb{E}[m_t] = \lambda_{1t}$. By Hoeffding's inequality,

$$\mathbb{P}\Big( \sum_{t=z_T}^{T} m_t - \sum_{t=z_T}^{T} \lambda_{1t} > a \Big) \leq \exp\Big( - \frac{2a^2}{T} \Big),$$

for any $a \geq 0$. Setting $\exp(-2a^2/T) = \delta$, we have $a = \sqrt{T\log(1/\delta)/2}$. Hence, with probability at least $1 - \delta$,

$$\sum_{t=z_T}^{T} m_t \leq \sum_{t=z_T}^{T} \lambda_{1t} + \sqrt{\frac{T}{2}\log\big(\frac{1}{\delta}\big)} \leq \sum_{t=1}^{T} \lambda_{1t} + \sqrt{\frac{T}{2}\log\big(\frac{1}{\delta}\big)}.$$

Since $\lambda_{1t} = \lambda_1 \sqrt{\log(dt)/t}$,

$$\sum_{t=1}^{T} \lambda_{1t} = \sum_{t=1}^{T} \lambda_1 \sqrt{\frac{\log(dt)}{t}} \leq \lambda_1 T \sqrt{\frac{1}{T}\sum_{t=1}^{T}\frac{\log(dT)}{t}} \leq \lambda_1 \sqrt{T}\sqrt{\log(dT)}\sqrt{1 + \log T},$$

where the first inequality is due to Jensen's inequality and the second is due to $\sum_{t=1}^{T}(1/t) \leq 1 + \int_{t=1}^{T}(1/t)dt = 1 + \log T$.

## B   Proof of Lemma 4.3

Recall that we defined $z_T = \max\Big( \frac{3}{C(\phi_1)^2}\log d, \frac{1}{C(\phi_1)^2}\log(\frac{T^2}{\delta'}) \Big)$ where $C(\phi_1) = \min(0.5, \frac{\phi_1^2}{256s_0})$. Due to assumptions A2 and A3 and Corollary 3.4, we have with probability at least $1 - \delta'$,

$$\hat{\Sigma}_t := \frac{1}{t}\sum_{\tau=1}^{t} \bar{b}(\tau)\bar{b}(\tau)^T \in C\Big( supp(\beta), \frac{\phi_1}{\sqrt{2}} \Big) \text{ for all } t \geq z_T, \tag{i}$$

where $\delta' < \delta$. It remains to show that when (i) holds, the left-hand side of (8) is smaller than $(\delta - \delta')/t^2$. In Section 3.2, we have shown that given the conditioning argument in (8) and the restriction (7) on $\pi_{a(t)}(t)$, $(\hat{r}(\tau) - \bar{b}(\tau)^T\beta|\mathcal{F}_{\tau-1})$ is $\tilde{R}$-sub-Gaussian for all $\tau = 1, \cdots, t$, with $\tilde{R} = O(1)$. Applying Lemma 3.2 with $(x_\tau, y_\tau) = (\bar{b}(\tau), \hat{r}(\tau))$ for $\tau = 1, \cdots, t$, $\phi = \phi_1/\sqrt{2}$, $R = \tilde{R}$ and $\delta = (\delta - \delta')/t^2$ completes the proof.

## C  Proof of Lemma 4.4

Suppose $t$ corresponds to subgroup (c-a). Since $a(t) = \underset{1 \leq i \leq N}{\operatorname{argmax}} \{b_i(t)^T \hat{\beta}(t-1)\}$ for this subgroup, $(b_{a(t)}(t) - b_{a^*(t)}(t))^T \hat{\beta}(t-1) \geq 0$ and

$$
\begin{aligned}
regret(t) &\leq regret(t) + (b_{a(t)}(t) - b_{a^*(t)}(t))^T \hat{\beta}(t-1) \\
&= (b_{a(t)}(t) - b_{a^*(t)}(t))^T (\hat{\beta}(t-1) - \beta) \\
&\leq ||b_{a(t)}(t) - b_{a^*(t)}(t)||_2 ||\hat{\beta}(t-1) - \beta||_2 \\
&\leq ||\hat{\beta}(t-1) - \beta||_2 \\
&\leq ||\hat{\beta}(t-1) - \beta||_1 \leq d_t,
\end{aligned}
$$

where the second inequality is due to Cauchy-Schwarz inequality and the third inequality is due to assumption A1. Hence, the sum of regrets from subgroup (c-a) is at most $\sum_{t=1}^{T} d_t$, which we can bound as follows.

$$
\begin{aligned}
\sum_{t=1}^{T} d_t &\leq \frac{\sqrt{128}}{\phi_1^2} s_0 \tilde{R} \sqrt{\log\left(\frac{edT^2}{\delta - \delta'}\right)} \sum_{t=1}^{T} \sqrt{\frac{1}{t}} \\
&\leq \frac{\sqrt{128}}{\phi_1^2} s_0 \tilde{R} \sqrt{\log\left(\frac{edT^2}{\delta - \delta'}\right)} T \sqrt{\frac{1}{T} \sum_{t=1}^{T} \frac{1}{t}} \leq \frac{\sqrt{128}}{\phi_1^2} s_0 \tilde{R} \sqrt{\log\left(\frac{edT^2}{\delta - \delta'}\right)} \sqrt{T} \sqrt{\log T}.
\end{aligned}
$$

The derivation is analogous to that of the upper bound of $\sum_t \lambda_{1t}$ because $d_t$ and $\lambda_{1t}$ have the same order in $t$. Meanwhile, we proved in Lemma 4.3 that the subgroup (c-b) is empty with probability at least $1 - \delta$. Therefore, with probability at least $1 - \delta$,

$$
R(T, c) \leq \sum_{t=1}^{T} d_t \leq \frac{\sqrt{128}}{\phi_1^2} s_0 \tilde{R} \sqrt{\log\left(\frac{edT^2}{\delta - \delta'}\right)} \sqrt{T} \sqrt{\log T}.
$$

## D  Proof of Theorem 4.1

Let $R(T, a)$ be the sum of regrets from group (a). Due to $R(T, a) \leq z_T$, Lemma 4.2, and Lemma 4.4, we have with probability at least $1 - 2\delta$,

$$
\begin{aligned}
R(T) &\leq R(T, a) + R(T, b) + R(T, c) \\
&\leq \max\left(\frac{3}{C(\phi_1)^2} \log d, \frac{1}{C(\phi_1)^2} \log\left(\frac{T^2}{\delta'}\right)\right) + \frac{\sqrt{128}}{\phi_1^2} s_0 \tilde{R} \sqrt{T} \log\left(\frac{edT^2}{\delta - \delta'}\right) + \lambda_1 \sqrt{T} \log(dT) \\
&\quad + \sqrt{\frac{T}{2} \log\left(\frac{1}{\delta}\right)} \\
&= O\left(s_0 \sqrt{T} \log(dT)\right).
\end{aligned}
$$