[Reviews · NeurIPS 2019]

Reviewer 1



interesting approach to contextual bandits by combining Lasso type methods with doubly-robust technique to handle missing data. The presentation of the paper needs some improvement, e..g, the relation between the sparse linear model used in Section 3 to the orginal bandit model (1) should be sharpened. It would be nice to put the obtained regret bounds into context of fundamental lower bounds, e.g. obtained in the context of adaptive compressed sensing E. Arias-Castro, E. J. Candes and M. A. Davenport, "On the Fundamental Limits of Adaptive Sensing," in IEEE Transactions on Information Theory, vol. 59, no. 1, pp. 472-481, Jan. 2013.

Reviewer 2



The exposition is quite unclear, and the paper seems hastily written. The main contributions of the paper is not outlined clearly, and not compared rigorously with existing results (see below). I believe there are enough theoretical contributions in the paper, but do not recommend publication as is. Ideally, I would recommend a "revise and resubmit". Even when presented in the best light, the main regret bound gives a significant deterioration in terms of its dependence on T, which is concerning. A thorough empirical evaluation is necessary to see if this deterioration is real, and whether it indeed scales better with dimension. 0) The role of double-robustness is quite unclear. It can act as a control variate and reduces variance, but a simple truncation would do the same thing. Is double-robustness fundamental to the regret bounds given by the authors? This point needs to be clarified, as the current manuscript seems to emphasize the doubly robust form very much. 0') Why is regressing on the doubly robust estimator \hat{r} better? Furthermore, the actual "double robustness" property never seems to appear in the paper, which is a bit concerning, and alarming. 1) The setting of paper (stochastic linear contextual bandits), and its main contributions should be clearly outlined much earlier. 2) The compatibility assumption required for Lasso seems to be used in the authors' analysis. The wording in S1 seems to imply that authors do not require this, which is misleading. 3) Bastani & Bayati studied a different setting where the contexts does not depend on arms. Therefore, a naive regret bound comparisons seems to be off. If we use a naive adaptation of authors' setting to theirs, then we get a O(s_0 log (dNT) \sqrt{T}) regret bound. This needs to be clarified in Section 1. 4) The substantially worse dependence on T needs to be clearly stated, and justified. Is it real? Is it from the analysis? Is the resulting improvement over N and s_0 worth the deterioration in practice? 5) Numerical experiments should prove/disprove the claims made in the above point (4). Authors also claim the fact their algorithm has one fewer hyperparameter than that of Bastani and Bayati as one of their main contributions. Does this actually lead to better practical algorithms? Minor comments: - S1, "competitive performance": What does this mean? Does it mean in terms of regret bounds? Or practical performance? - S1, Li et al (2010): what was the dimension here? - S1, "reinforcement learning": replacing this with bandit settings would strengthen the statement, although this whole paragraph seems misleading at best. - S1, "not utilized by previous methods": does this refer to previous contextual bandit algorithms? The wording is quite confusing, and depending on what the authors meant, I would disagree.

Reviewer 3



The authors propose a double robust bandit algorithm for solving the contextual linear bandit problem. In particular, their algorithm is independent of N, the number of arms and instead depends on the sparsity of the context. This is achieved by dipping into the missing data literature and adapting the doubly robust technique (Bang and Robbins 2005). Experimental results are shown on synthetic datasets with varying correlation patterns between the context vectors. Comments: Overall, the paper is well-motivated and written in a clear manner. Contextual settings are of vital interest in many real-world problems such as advertising and recommender systems. Reducing the dependence of regret to the sparsity of the context vector is of immense value since side information is usually very high-dimensional and sparse.

[Author Response · NeurIPS 2019]

We thank the reviewers for their helpful comments and suggestions. Due to space limitation, we focus on 1) role of doubly-robustness, 2) comparison with Bastani and Bayati, 3) hyperparameters and 4) real-world experiments.

**Role of Doubly-Robustness (DR):** A key strength of doubly-robust (DR) method is to obtain an estimate of the reward corresponding to the average context $\bar{b}(t)$. Since $\bar{b}(t)$ satisfies the compatibility condition, we can make use of the fast convergence of the Lasso estimator. If the reward for every arm is observed, the problem becomes much easier. However in bandits, only $r_{a(t)}(t)$ is observed and the rewards of other arms remain missing. In missing data literature, there are two approaches: inverse probability weighting (IPW) and imputing (IMP). IPW and IMP require correct specification of the probability of observation and the imputation model, respectively. DR method uses both auxiliary models, but the consistency of the estimator is guaranteed when either of the models is correctly specified. **In the bandit setting, the probability of observation is known, thus both IPW and DR yield consistent and unbiased estimators.** IPW was used in the EXP4.P bandit algorithm of Beygelzeimer et al. (2011). **Another strength of DR is that when both models are correctly specified, the resulting estimator has the minimum variance.** This efficiency is obtained by projecting the IPW estimating function on to the tangent space spanned by nuisance parameters in the probability of observation (or allocation in bandit setting). The form of $\hat{r}$ reflects this adjustment from an IPW form. We show below that our DR estimator $\hat{r}(t)$ guarantees efficiency gain over the IPW estimator. Let $\tilde{r}(t)$ denote the IPW esimator of the reward corresponding to $\bar{b}(t)$ and let $\hat{r}(t)$ be the DR estimator as defined in the text.

Hence, $\tilde{r}(t) = \frac{r_{a(t)}(t)}{N\pi_{a(t)}(t)}$ and $\hat{r}(t) = \tilde{r}(t) + \bar{b}(t)^T\hat{\beta}(t-1) - \frac{b_{a(t)}(t)^T\hat{\beta}(t-1)}{N\pi_{a(t)}(t)}$. The variance of $\tilde{r}(t)$ given filtration $\mathcal{F}_{t-1}$

is $\mathbb{E}\left[\left\{\frac{\eta_{a(t)}(t)}{N\pi_{a(t)}(t)} + \frac{b_{a(t)}(t)^T\beta}{N\pi_{a(t)}(t)} - \bar{b}(t)^T\beta\right\}^2 \Big| \mathcal{F}_{t-1}\right]$. Due to Assumption 4 on the sub-gaussian error $\eta$, the first term $\eta_{a(t)}(t)/N\pi_{a(t)}(t)$ is bounded by a constant when $\pi_{a(t)}(t) \geq O\left(\frac{1}{N}\sqrt{(\log d + \log t)/t}\right)$. However, the second term $\frac{b_{a(t)}(t)^T\beta}{N\pi_{a(t)}(t)}$ can still be large. Constant variance is important because the variance ($\tilde{R}^2$ in the text) appears in the regret bound in Theorem 4.1. To achieve a constant variance, we need $\pi_{a(t)}(t)$ be larger than a predetermined constant value, $\frac{1}{N}p_{min}$. **Simply truncating the value will produce a biased estimate when $\pi_{a(t)}(t)$ is actually smaller than $\frac{1}{N}p_{min}$, and the Lasso property (Lemma 3.2) will not hold either.** If we instead directly constrain $\pi_{a(t)}(t)$ to be larger than $\frac{1}{N}p_{min}$, this will lead to suboptimal choices of arms and Theorem 4.1 will not hold. In contrast, the variance of $\hat{r}(t)$ is $\mathbb{E}\left[\left\{\frac{\eta_{a(t)}(t)}{N\pi_{a(t)}(t)} + \frac{b_{a(t)}(t)^T\beta^*}{N\pi_{a(t)}(t)} - \bar{b}(t)^T\beta^*\right\}^2 \Big| \mathcal{F}_{t-1}\right]$, where $\beta^* = (\beta - \hat{\beta}(t-1))$. Now, we have a constant variance under $\pi_{a(t)}(t) \geq O\left(\frac{1}{N}\sqrt{(\log d + \log t)/t}\right)$ with high-probability due to the fact $||\beta^*||_1 \leq O\left(\sqrt{(\log d + \log t)/t}\right)$ with high-probability ($\because$ Lemma 3.2 on observations until $t-1$). The regret bound in Theorem 4.1 holds under (i) $\pi_{a(t)}(t) \geq O\left(\frac{1}{N}\sqrt{(\log d + \log t)/t}\right)$ but not under (ii) $\pi_{a(t)}(t) \geq \frac{1}{N}p_{min}$. (We skipped details on this part, but the relevant part is the bound on $\sum_{t=z_T}^{T} m_t$ in Section 4.1.) Also note that the restriction (i) is much weaker than (ii) since the term $\sqrt{(\log d + \log t)/t}$ converges to 0 as $t$ increases, inducing exploration in early stages and greedy choices when $t$ is large.

**Comparison with Bastani and Bayati (BB):** As Reviewer 2 mentioned, **our method and BB deal with different settings and direct comparison may not be meaningful. There are no previous works dealing with our setting of large $N$.** Having said, we agree with Reviewer 2 that if we use our method **in the setting of BB, when $N$ is not that large**, it will produce a regret of order $O(s_0 \log(dNT)\sqrt{T})$ which is larger than that of BB. We stress that our method and the method of BB are designed for different settings. BB guarantees the best performance when number of arms is moderate, and when we can assume that each arm has different regression parameter. In this case, BB has an advantage. However, there are cases where the number of arms is very large. Especially, when we recommend a news article or a shopping item, or when we place an advertisement on the web page, the number of possible action selections is very large. Moreover, the lists of news articles, shopping items, or advertisements change day by day (even change a lot in a single day). In this case, it would not be feasible to assign a different parameter for every new incoming item, and also to conduct forced-sampling of arms according to a predetermined schedule. Therefore, in cases where the number of arms is large and the arm set changes with time, our method will show advantage over BB.

**Hyperparameters:** In online learning, it is difficult to simultaneously tune the hyperparameters and achieve high reward, and it is crucial to have a smaller number of hyperparameters. Due to difficulty in simultaneous tuning and optimization, at the beginning rounds, we should sacrifice learning the tuning the hyperparamters. In this stage, the accumulation of rewards remains slow because we do not know yet which values of hyperparameters are best suited to our algorithm. When we tune the values by grid search, then the amount of time required for tuning is exponential in the number of tuning parameters. Therefore, having one less tuning parameter can result in much short time required for tuning process.

**Real data experiments:** We have some encouraging real data example using the YAHOO news article recommendation log data. We will include the results in the supplementary material in the future.

[Meta-Review · NeurIPS 2019]

The reviewers agree that the paper provides an interesting technique to an important problem. There were some concerns around clarity, but we believe these issues can be fixed towards a camera ready version. I urge the authors to revise the paper based on the provided reviews.